# Diversity and Domestication Status of Spider Plant (*Gynandropsis gynandra*, L.) amongst Sociolinguistic Groups of Northern Namibia

**Barthlomew Chataika [1],\*** , **Levi Akundabweni [1]**, **Enoch G. Achigan-Dako [2]**, **Julia Sibiya [3]**, **Kingdom Kwapata [4] and Benisiu Thomas [1]**

[1]  Crop Science Department, Faculty of Agriculture and Natural Resources, University of Namibia, Ogongo Campus, Private Bag 5520, Oshakati, Namibia; lakundabweni@unam.na (L.A.); bthomas@unam.na (B.T.)

[2]  Laboratory of Genetics, Horticulture and Seed Science, Faculty of Agronomic Sciences, University of Abomey-Calavi, Cotonou 999105, Benin; enoch.achigandako@uac.bj

[3]  School of Agricultural, Earth and Environmental Sciences, University of KwaZulu-Natal, P.Bag X01, Scottsville, Pietermaritzburg 3209, South Africa; sibiyaj@ukzn.ac.za

[4]  Department of Horticulture, Lilongwe University of Agriculture and Natural Resources, P.O. Box 219, Lilongwe, Malawi; kwapata@yahoo.com

\*  Correspondence: barthchataika@gmail.com

**Abstract:** Knowledge on the diversity and domestication levels of the spider plant (*Gynandropsis gynandra*) has the potential to affect pre-breeding for client-preferred traits, yet information is scarce in Namibia due to limited research. We investigated indigenous knowledge on the species diversity and domestication levels in the regions of Kavango West, Ohangwena, Omusati, Oshana, and Oshikoto of northern Namibia. Semi-structured interviews involving 100 randomly selected farming households, four key informant interviews, and a focus group discussion were conducted. Descriptive and chi-square tests were conducted using IBM SPSS version 20. Out of the possible four morphotypes, the results suggested that only one with green stem and green petiole existed and was associated with soils rich in organic manure. Spider plant abundance was reported to be on the decline, due to declining soil fertility. On a scale of 0 (wild species) to 6 (highest level of domestication), an index of 1.56 was found and this implied very low domestication levels. Furthermore, the study found significant differences in the trends of domestication across the sociolinguistic groups ($\chi^2$ (12, N = 98) = 46.9, $p < 0.001$) and regions studied ($\chi^2$ (12, N = 100) = 47.8, $p < 0.001$), suggesting cultural and geographical influences. In conclusion, the findings constituted an important precedent for guiding subsequent pre-breeding efforts.

**Keywords:** pre-breeding; morphotypes; domestication index; indigenous knowledge; sociolinguistic groups; client-preferred traits

## 1. Introduction

In many parts of sub-Saharan Africa (SSA), the spider plant (*G. gynandra* L. (Briq.) is an important indigenous vegetable that is neglected and underutilized but plays a crucial role in food and nutrition security and income generation of the rural poor [1–4]. It grows as a volunteer weedy crop in farmers' fields and in the wild during the rainy season [5–7]. Depending on the knowledge of the farmers, the vegetable is either removed as a weed or spared so that it can be harvested for use as a vegetable / relish or for sale in local markets.

No studies, however, have been reported on genetic diversity and domestication trends of the species, despite the emerging shift aimed at integrating indigenous and neglected vegetables in smallholder farming systems. The lack of attention means that the potential value of the spider plant remains underestimated and underexploited. This article reports on a study conducted in northern Namibia to assess the potential for domestication of the spider plant and promotion of its cultivation. The plant is well known in northern Namibia amongst different sociolinguistic groups. Farmers harvest the green leaves when they are abundant in the rainy season between November and March. The harvested leaves are either cooked as a relish, sold in the market as green or processed vegetable. Processing is done to preserve the vegetable for use in the dry season. Farmers preserve spider plants either by sun drying or bleaching followed by drying or forming of pellets which are used as a relish, sold on the open markets [6] and used during some of the traditional practices such as "Olufuko" amongst the Oshiwambo sociolinguistic groups [8]. (Source of information???). The cultural and economic importance of spider plant amongst sociolinguistic groups of northern Namibia calls for the need to promote its production which can be achieved by utilizing its genetic diversity.

Plant domestication and genetic improvement can be enhanced through the utilization of genetic diversity [9]. Diversity refers to the number of morphotypes or accessions that are found and used in a particular region [10] resulting from culture, traditional knowledge, the introduction of new species, domestication and crop improvement by farmers.

In addition, our interest in plant domestication was based on whether, in northern Namibia, there was a process of plant population evolution [11], leading to genetic change emanating from exploitation, selection, cultivation of the selected wild plants, and adaption to the agroecosystems and the human needs [12]. Such diversity and domestication, if found, would provide an opportunity to identify the 'elite' species with desirable utilization attributes such as nutritional and medicinal traits for further propagation to serve the rural residents.

These research interests were based on previous recommendations of this nature amongst ethnobotanists in Africa. For example, Dansi et al. [1] and Sogbohossou et al. [13] recommended an ethnobotanical investigation to evaluate, identify, document and prioritize interventions for reducing production constraints, improving agricultural practices and assessing the species contribution to household income.

## 2. Materials and Methods

### 2.1. Research Sites Sampling Design and Research Tools

The study was conducted in Kavango West, Ohangwena Omusati, Oshana, and Oshikoto regions of Northern Namibia between June and July, 2018. Namibia is located in south western part of Africa, surrounded by Angola to the north, Botswana to the east, South Africa to the south and Atlantic Ocean to the west (Figure 1).

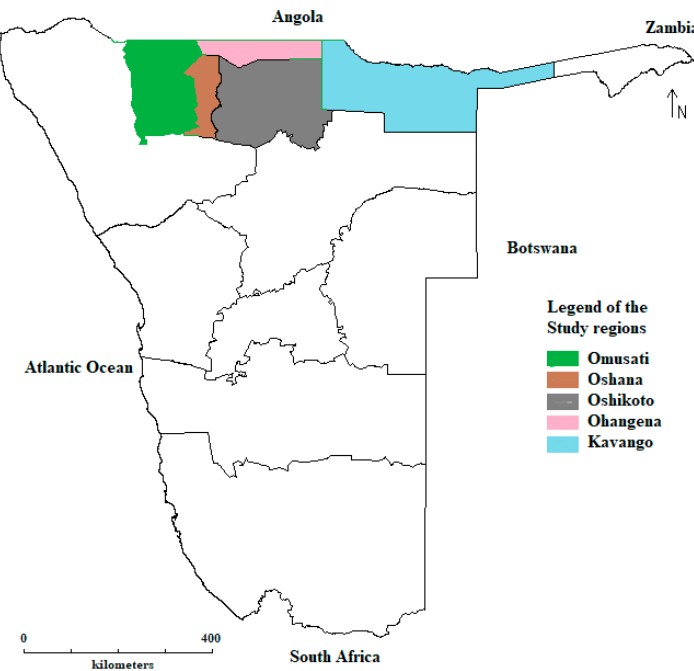

**Figure 1.** Map of Namibia showing the five study regions.

Previous studies have shown that the spider plant grows naturally and mostly in north central and north eastern Namibia [14]. The plant is known by several names as it is a familiar plant to several different sociolinguistic groups in that region as: Cat's whiskers or Spider flower (English), Ombidi (Oshikwanyama), Omboga (Oshindonga), Ombowa yozongombe or Ombowayozondu (Herero) and Gomabeb (Damara) [14].

The study used three tools to capture the diversity and domestication levels of spider plant in Northern Namibia. These tools were semi structured interviews targeting farming households; focus group discussion (FGD) which involved a farmer group and key informant interviews (KII) targeting agricultural extension staff in the Ministry of Agriculture.

Sampling was done at three stages. Firstly, purposeful selection of the four regions was done based on the outcome of the preliminary field survey and literature review. The second stage was random selection of constituencies in each of the regions. Constituencies are local administrative areas that form a region. One constituency was selected in each of the five study regions. Finally, a sample of 100 farming households was randomly drawn from the five regions (20 farming households per sampled constituency in each of the regions) for semi-structured interviews. Cochran's sample size formula ( $n0 = \left(z^2pq\right)/e^2$.) was used to determine the minimum sample size [15] based on 8% margin of error (*e*) and 95% confidence interval (z-value = 1.96). An estimate of the population which had knowledge of spider plant (*p*) was set at 80% based on the preliminary assessment. The choice of the margin of error took into consideration the resource constraints, while ensuring that it fell in between 10% and 5%. Using the formula, a sample size of 96 farming households was found and this was adjusted upwards to 100 households. In order to account for non-responses, the sample size was adjusted upwards by 10% per constituency and only the first 20 households were interviewed (Table 1). The interviews targeted the head of the household or the spouse.

**Table 1.** Study constituencies and number of sampled farming households per region.

| Region | Constituency | Number of Households | * Percentage of Farming Households | Farming Households Sampled |
|---|---|---|---|---|
| Omusati | Anamulenge | 2500 | 53 | 20 |
| Oshana | Okatana | 2600 | 12 | 20 |
| Ohangwena | Engela | 4900 | 36 | 20 |
| Oshikoto | Omuntele | 3300 | 33 | 20 |
| Kavango West | Kapako | 4200 | 31 | 20 |
| Overall | | 17500 | 33 | 100 |

* Percentage of farming households is based on constituency estimates as per the national demographic survey of 2016.

In addition, one farmer group from Ohangwena region, and four agricultural extension staff were sampled for FGDs and as key informants respectively, to triangulate the findings. The farmers group was a formal cooperative formed by the Ministry of Agriculture, Water and Forestry through the Japan's International Cooperation Agency (JICA) project to promote crop production using conservation agriculture. The farming households were mobilized and sampled with assistance from the Agricultural extension staff based at the regional and constituency offices. The interviews for the sampled households were conducted at their homesteads after explaining the objectives of the study and getting their consent. Demographic characteristics and ethnicity of the sampled households were compiled.

Three key research questions were asked during the interviews with research participants. The first question was for the respondents to describe the types of spider plant found in their areas. The description included the colour, hairiness, local names and other agro-morphological features to determine the diversity of the species. The respondents were then asked to describe the agro-ecological niche where spider plant was found growing in abundance, including associated cropping systems and further to explain how they managed the spider plant when it grew either in the wild or in their fields. This was an open-ended question, and thus based on the explanation given, a score was assigned to indicate levels of domestication. We used a model of seven scores [16] as follows: Level 0: Wildlife species; Level 1: species just spared in the fields during field works; Level 2: Species spared in the fields but benefit from some care for its growth; Level 3: species transplanted from nature to the cultivated fields or home gardens; Level 4: Species well cultivated and reproduced; Level 5: Species cultivated with some selection activities; Level 6: Pests and diseases are known as well as their means of control. Finally, the respondents were asked to determine domestication trends of spider plant over the years and the underlying factors.

*2.2. Data Analysis*

Data from household interviews were subjected to descriptive and chi-square tests using IBM SPSS Statistics for Windows, Version 20.0. Armonk, NY: IBM Corp. Depending on the type of data, either means or frequencies were generated and presented in tables and graphs. A Shapiro-Wilk's test ($p > 0.05$) [17] and measures of skewness and kurtosis z-values were used to test normality of data on abundance of spider plant and associated cropping systems. In addition, tests for equality of variances were done using non-parametric Levene test ($p > 0.05$) [18]. Qualitative data from FGDs and key informant interviews were summarized and analyzed based on themes [19,20].

**3. Results**

*3.1. Demographic Characteristics of the Interviewed Households*

In this study, married respondents comprised 44%, while the single (never married or were divorced) constituted 49% and the remaining six percent were widows. The percentage of farming households within each constituency that were sampled, from the seven sociolinguistic groups, ranged from 7% for Chokwe to 22% for Kwanyama (Table 2). The average number of years in formal school

ranged from one year for Chokwe sociolinguistic group to 8.7 years for Kwambi (Table 2) with an overall mean of 6.6 years.

**Table 2.** Proportion of interviewed farming households, their education levels and mean ages across the sociolinguistic groups and regions of northern Namibia.

| Category | | Proportion of Respondents (%) | Education Level (Years) | Mean Age (Years) |
|---|---|---|---|---|
| **Sociolinguistic group** | Chokwe | 7 | 1.0 | 46.9 |
| | Kwambi | 20 | 8.7 | 50.4 |
| | Kwangali | 13 | 5.6 | 34.0 |
| | Kwanyama | 22 | 6.2 | 55.7 |
| | Mbadja | 9 | 8.4 | 44.8 |
| | Mbalantu | 11 | 6.2 | 55.7 |
| | Ndonga | 18 | 7.4 | 53.2 |
| **Regions** | Kavango | 20 | 4.0 | 38.5 |
| | Ohangwena | 21 | 6.0 | 56.4 |
| | Omusati | 20 | 7.2 | 50.8 |
| | Oshana | 20 | 8.7 | 51.4 |
| | Oshikoto | 19 | 7.7 | 52.5 |
| | Overall | 100 | 6.6 | 50.0 |

The mean age of the farming households was 50.0 years, with a standard error of 1.7. Kwangali had the youngest group of respondents (34.0 years) while Kwanyama and Mbalantu had the oldest, each with an average of 55.7 years (Table 2). In terms of regions, Kavango had relatively younger participants (38.5 years) while Ohangwena had on average older participants (56.4 years) in the survey (Table 2). The primary occupation of the respondents was farming (90%) followed by business (7%), scholars (2%) and finally formal employment (1%).

### 3.2. Diversity of Spider Plant

In literature, the spider plant is categorized into four morphotypes based on pigmentation, and these are green, purple, pink and violet [21]. These four morphotypes can either be globulous or have hairs with varying density. In this study, plants with green stems and green petioles (Figure 2a,b) were the only morphotypes of the spider plant reported to exist in the five regions. No respondent mentioned having seen or used purples spider plant morphotypes (Figure 2c,d).

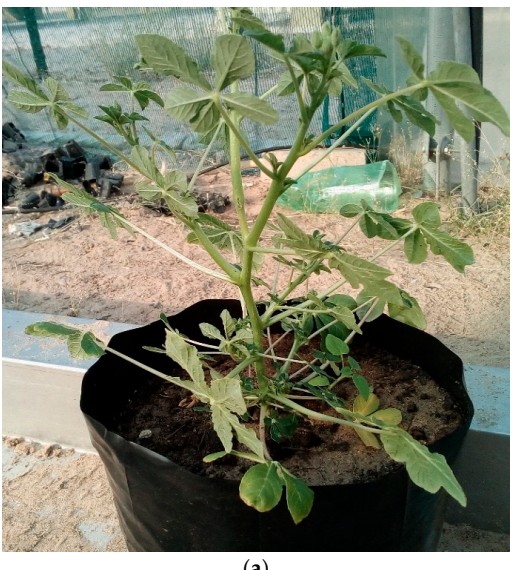 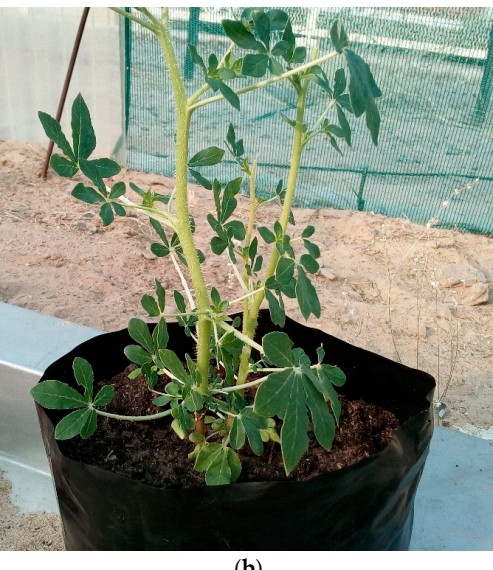

(**a**)　　　　　　　　　　　　　　　　　　　　　　　　　　　　　　　(**b**)

**Figure 2.** *Cont.*

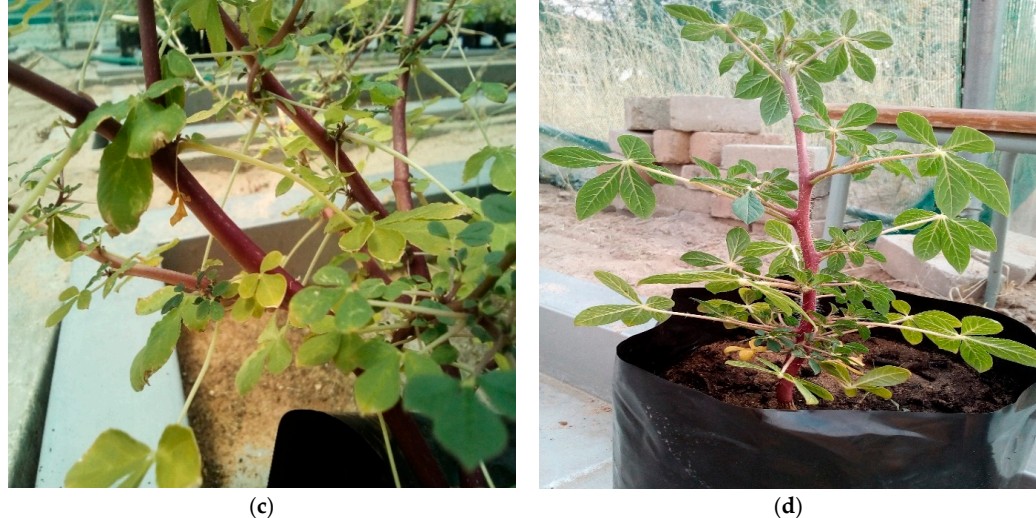

(**c**)    (**d**)

**Figure 2.** Some of the spider plant morphotypes, (**a**) green stem without trichomes, (**b**) green stem with trichomes, (**c**) purple stem without trichomes, and (**d**) purple stem with trichomes.

In terms of hairiness, 48% reported having seen and used globulous spider plants while 52% reported knowing spider plants with dense trichomes or hairs (Table 3). Most respondents from Omusati and Kavango regions reported having seen spider plant with dense trichomes while while the majority of the respondents from Oshana and Ohangwena reported that they saw and used globulous spider plant.

**Table 3.** Percentage of respondent reporting the hairiness of the green spider plant across the study regions.

| Region | Globulous | Dense Hair |
| --- | --- | --- |
| Omusati (n = 20) | 15 | 85 |
| Oshana (n = 20) | 80 | 20 |
| Ohangwena (n = 21) | 71 | 29 |
| Oshikoto (n = 19) | 68 | 32 |
| Kavango (n = 18) | 6 | 94 |
| Overall (n = 98) | 48 | 52 |

The spider plant was locally called Ombidi (n = 42) or Omboga (n = 38) in Omusati, Oshana, Ohangwena and Oshikoto regions while in West Kavango (n = 20) it was known as Mpungu. Abundance of spider plants in the wild, in crop fields and homestead was reported to be associated with soils of high organic manure ($p < 0.001$) particularly cow dung (75%) (Figure 3). The study also found that the abundance of spider plant was not confined to a specific cropping system ($p < 0.001$) (60% of the respondents) (Figure 3). Normality tests for both abundance and associated cropping systems using skewness and kurtosis produced the z-values in between −1.96 and 1.96 suggesting normal distribution.

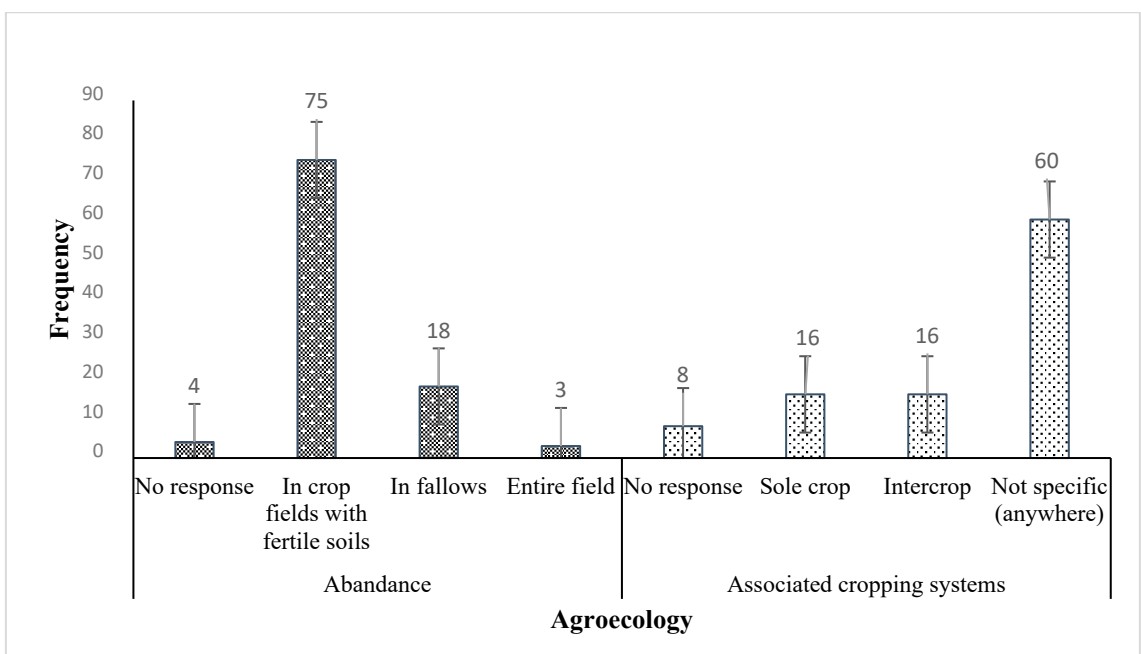

**Figure 3.** Frequency of respondents indicating abundance of spider plant and associated cropping systems of northern Namibia.

The respondents reported that the green form of spider plant was only found during the rainy season. Farmers harvest the green leaves when they are abundant in the rainy season and dry them either as pellets or bleached vegetable to preserve for the dry season.

*3.3. Domestication Levels and Trends*

The results showed that there was some domestication of spider plants taking place in the study areas. About 46% of the farmers did not remove the spider plant when working in the fields but allowed it to grow for harvesting, and the other 54% took some care to enable the spider plant to grow properly. The two scenarios corresponded to levels one and two of domestication, and the scores translate to a domestication level of 1.54 ± 0.501. This implied that the species was spared in the fields during field work and more often benefitted from some care for its growth. There were no reports of active domestication by farmers such as transplanting spider plants from the wild into fields and gardens, deliberate cultivation, including the selection of seeds from particular plants for reproduction, and use of means to control pests and diseases.

There was variation in the level of attention paid by farmers from different sociolinguistic groups. All farmers amongst the Chokwe and Kwangali groups provided some care to promote growth of the plant while the Mbalantu group just spared the crop during field work as illustrated in Figure 4 below.

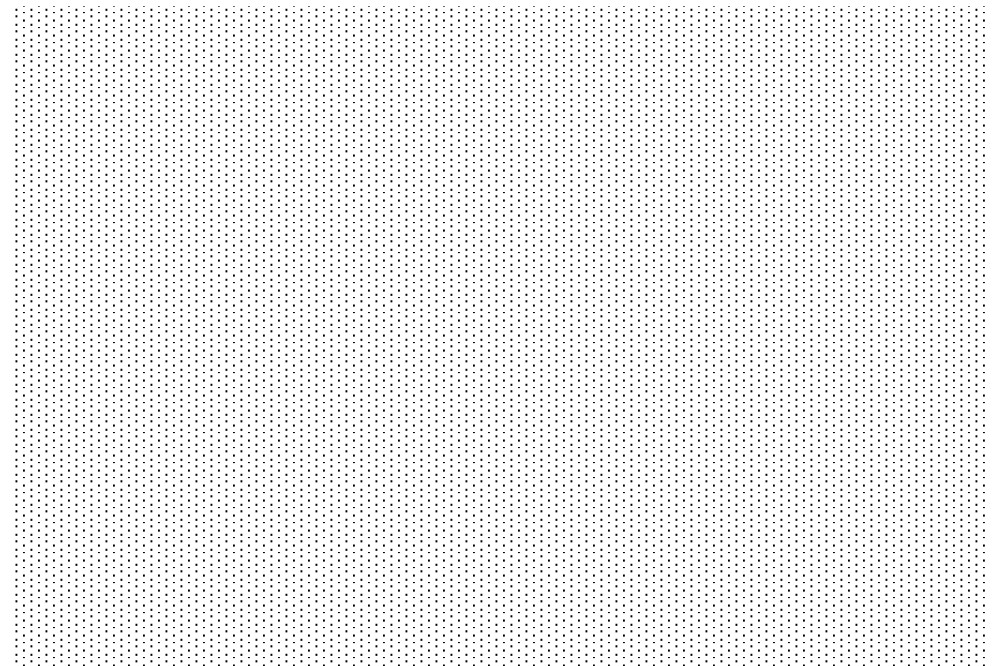

**Figure 4.** Levels of domestication of spider plant across different sociolinguistic groups of northern Namibia.

Trends of domestication significantly differed across the sociolinguistic groups ($\chi^2$ (12, N = 98) = 46.9. $p < 0.001$) (Figure 5) and regions ($\chi^2$ (12, N=100) = 47.8, $p < 0.001$) (Figure 6). In general, a decline trend in domestication was reported across the regions except in Kavango west where the respondents reported that the trend was going upwards ($p < 0.001$). Chokwe and Kwangali sociolinguistic groups reported increasing trends while the rest of the sociolinguistic groups, which were from Omusati, Oshana, Ohangwena and Oshikoto, reported a general downward trend. There was the correlation in domestication trends between the location of sociolinguistic group and the regions.

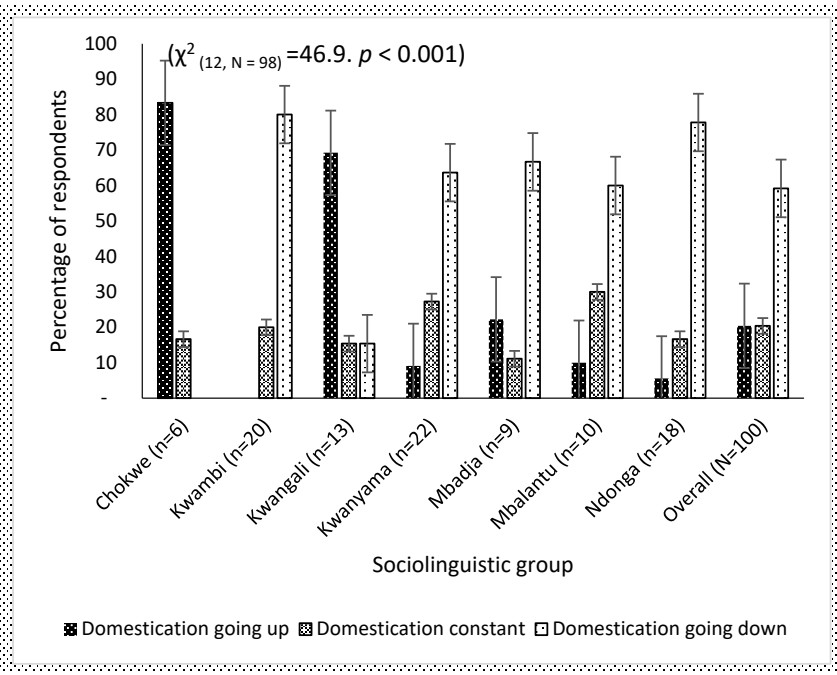

**Figure 5.** Trends of domestication of the spider plant across the different sociolinguistic groups of Northern Namibia.

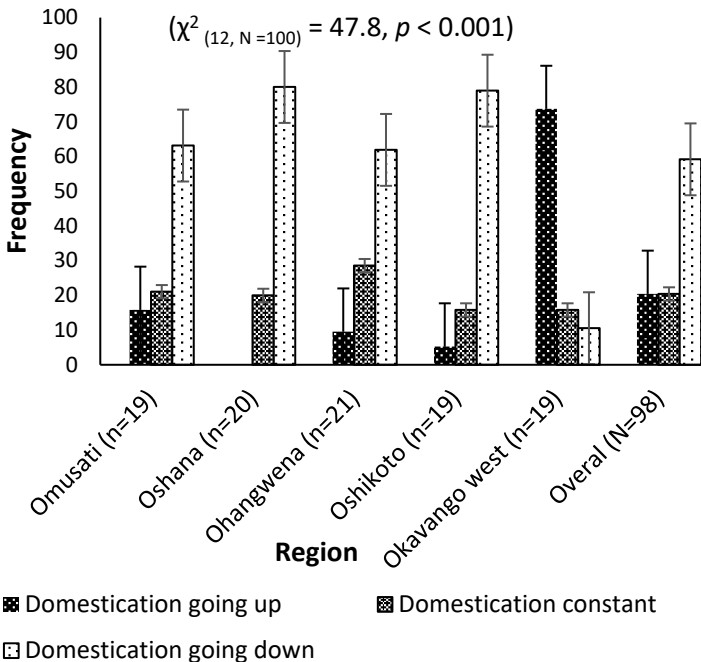

$$(\chi^2_{(12,\ N=100)} = 47.8,\ p < 0.001)$$

**Figure 6.** Trends of domestication of the spider plant across the regions of Northern Namibia.

The main reason for the downward trend was reported to be due to declining soil fertility, as the species was believed to be associated with fertile soils (Table 4). Poor rainfall, associated with drought conditions was the other factor leading to the declining trends in the domestication of spider plant.

**Table 4.** Reasons given by the respondents for the observed trends in spider plant domestication.

| Trend of Domestication | Reasons for the Trend | Frequency |
|---|---|---|
| | **Non Responses** | **38** |
| Downward trends | Drought/poor rainfall | 13 |
| | Infertile soil | 19 |
| | Water lodging/high rainfall | 1 |
| Upward trends | Fertile soil | 18 |
| | Good rainfall | 4 |
| | Seeds are broadcast or left to dry in the field | 4 |
| | Adapt to many environments | 3 |

## 4. Discussion

### 4.1. Diversity of Spider Plant in Northern Namibia

The study showed a lack of genetic diversity and ecological threats to the use of what is regarded as a useful crop. However, in view of the link between cow manure and spider plant growth alongside widespread cattle keeping across the regions, and adaptation to different cropping systems, there is an opportunity for agricultural extension investment and assistance to farmers to grow spider plant.

Genetic diversity has been reported to be the key to achieving the gains in production and productivity [9] through the enhancement of plant domestication and genetic improvement. Assessment of spider plant diversity in this study was based on stem and petiole pigmentation, and trichomes density. The study showed that only green stem, green petiole morphotypes, with and without trichomes existed in the study region.

The findings on pigmentation might imply limited opportunities for identifying superior accessions with the desired traits for use in genetic improvement. In addition, the results may also reflect the levels

of knowledge of the species amongst the farmers, which might be limited by the fact that the vegetable was considered as a wild crop which did not undergo exploitation through domestication. Above all, pigmentation is just one of the several morphological characters which are used to distinguish genetically diverse accession. The findings of this study contrasted other studies which reported existence of four morphotypes in countries such as Kenya and Benin [21,22] and a mixture of green and purple colours in farmers' cultivars in Kenya [23]. In Namibia, it was reported that spider plant grew naturally particularly in Oshikoto, Oshana, Omusati, Ohangwena, Kavango East and West, Kunene and Zambezi regions but did not report the diversity of the morphotypes [14].

In contrast with green accessions, purple accessions are reported to be associated with higher nutrient density [23] which have health benefits, and also contain phytochemicals which confer resistance to insect pests [24]. The current diversity status in the study regions, therefore, might imply limited options in accessing the diversity of nutraceutical benefits from spider plant amongst the sociolinguistic groups. This limited diversity could also have a negative impact in designing and consequent implementation of breeding programs that seek to maximize genetic diversity with the aim of breeding ideal varieties, with desirable traits. Nonetheless, there is a possibility of the existence of the other morphotypes but this could only be confirmed by following up this study with field collection, identification and genetic characterization of the species across the five regions. Farmers' knowledge might only be limited to the green petiole, green stem morphotypes.

The second morphological character reported was the existence of spider plants with and without trichomes. Trichomes play important roles in pest resistance through either physical obstruction or production of phytochemicals which are toxic to herbivores [25]. The phytochemicals produced in the trichomes, as reported in plants such as *Plectranthus ornatus* [26] are also reported to impact on nutritional value of the plant species. The existence of spider plants with trichomes, therefore, offers an opportunity for identifying accessions with pest resistance potential and consequently enhance domestication.

Thirdly, the study identified a strong association of spider plant with organic matter, particularly cow manure. This was consistent with the findings of other researchers [14] who reported abundance of spider plants where manure or household refuse accumulated. This offers an opportunity in designing best-fit agronomic practices that would maximize the abundance and production as one of the ways of stimulating domestication of the species. Farmers in the study regions keep a lot of cattle, as such it should be easy to produce enough cattle manure which could be used for the production of spider plant. Furthermore, spider plant abundance was found to be widely spread across different cropping systems thus giving a wide choice for farmers to integrate spider plant production in different cropping systems. This also suggested that the spider plant did not compete negatively with associated crops in the different cropping arrangements. This finding, however needs to be followed by controlled agronomic studies to quantify the interactions of different cropping arrangements with spider plant and establish the optimum organic manure quantity and associated nutrient levels that would maximize production.

### 4.2. Domestication of Spider Plant

Domestication and scaling-up the adoption of indigenous vegetable species with high nutraceutical potential is one of the promising strategies that countries, that want to generate and sustain broad-based wealth, need to embrace. Understanding of domestication syndrome is considered as the starting point for the developing new and orphan crops [27]. The finding of this study suggested that spider plant was at early stages of domestication (level 2) and this was consistent with the finding of other researchers [28] who found that spider plant at level two of domestication in Gbede village in Benin. This implied that the vegetable remains neglected and thus calling for prompt action owing to its importance. One of the ways of accelerating domestication is through the use of best-fit agronomic practices that improve growth and leaf yield [29]. Generally, domestication is considered to reduce the genetic diversity hence creating a domestication bottleneck [30]. According to the theory of evolution low genetic diversity is believed to expose the species to the increased risk of extinction. This is

because of the narrow genetic base which limits the capability of the species to survive changing agro-climatic conditions. Researchers have identified some of the genes underlying domestication and diversification [31] which can be exploited to adequately understand the domestication trends and diversity of spider plant at genetic level. In this study, both diversity and domestication levels were low probably because only farmers' responses were used and these could be limited to their level of knowledge.

Because of the perceived knowledge gap, it is proposed that follow-up studies should be done during the rainy season to collect and identify accessions growing both on farmers' fields and in the wild. The proposed study would embrace participatory domestication techniques [32] covering a range of species that have the potential of meeting diverse market requirements and domestic needs including nutritional and medicinal qualities. In addition, the proposed study would integrate innovative domestication with processing and commercialization as proposed by Leakey & Asaah [33] in order to make progress in the cultivation of the underutilized species. It was anticipated that the study would add more information on the available morphotypes and open up opportunities for further breeding activities.

*4.3. Implications for Research and Development*

This research has generated a wealth of knowledge for modelling future research aimed at promoting the domestication of spider plant. In order to develop breeding program for orphan crops such as spider plant, cultivar development [34] is key and this is dependent of diversity of the genetic materials, the potential of the material to adapt to wide range of environmental bottleneck and willingness of the farmers to domesticate. Therefore, the current diversity and domestication continuum of spider plant in Namibia provide opportunities for devising ideal genetic enhancement approaches for supporting de novo domestication. We found low diversity and downward trends of domestication, implying that the species is at the risk of genetic erosion. Since the trends in domestication were different across different sociolinguistic groups, identification of the underlying cultural factors need to be investigated. Furthermore, trends in domestication of spider plant will considerably depend on the benefits emanating from its use and also the mode of harvesting amongst other reasons. For example, uprooting the whole plant before flowering prevents seed dispersal hence leads to reduced diversity of the species. A participatory study on the appropriate methods of harvesting and utilization amongst different sociolinguistic groups is therefore recommended. We further recommend investigation on modes of conservation and production technologies to enhance domestication of the species. Use of participatory domestication methods to genetically improve orphan crops [35] promises to be a useful approach. At policy level, the roles indigenous vegetables such as spider plant, play in complimenting the major crops need to be recognized and mainstreamed in the planning framework to promote domestication. In Namibia, horticultural research only started in 1995 but did not include indigenous vegetables. The main focus was to test the suitability of varieties from South Africa in Namibian agro ecologies [36].

The proposed approaches will serve as models for designing and implementing research and development of ideal genotypes that respond to the needs of clients. Breeding efforts for orphaned vegetables are still at their infancy levels. The limited diversity of the species implies limited options for identifying candidate accessions for genetic improvement, as such researchers may need to widen the sources of breeding materials from other areas. Nevertheless, the positive association of spider plant with fertile soils offers both an opportunity and a challenge. Namibia is generally dry with poor soils which do not support optimum production of spider plant. However, Namibian farmers rear a lot of livestock which produce manure which can be used in the production of spider plant hence enhancing its domestication. Residues from spider plant can also potentially be used as feed for livestock. Finally creating awareness on the importance of domesticating spider plant through trainings and information campaigns will potentially enhance the use of spider plant and consequently its domestication.

## 5. Conclusions

In northern Namibia, diversity of the species was limited to green petiole and green stem morphotypes but this needs to be confirmed with a follow-up study which should aim at collecting, curating, identifying and characterizing the species and its wild relatives. The species are still in the early stages of domestication and the abundance was reported to be decreasing due declining soil fertility, hence calling for urgent action. In-depth understanding of cultural influence towards utilization of the species, including identification of preferred traits and production constraints constitute key pre-breeding stimulants aimed at popularizing the domestication of the Spider plant. In this study, we identified possible threats to the growth of the plant, which are key to informing further research to investigate options to improve domestication of the plant in northern Namibia. This constitutes a key step towards improving knowledge of the spider plant and its utility as a food crop.

**Author Contributions:** B.C. conceptualized the research design and the methodological approach, conducted the investigation, curated, and analyzed data, and wrote the original draft. L.A. and E.G.A.-D. reviewed the methodology and results, supervised the research work and edited the manuscript. J.S. and K.K. supervised, reviewed and edited the manuscript B.T. proofread, edited, and provided support with the ethical clearances and logistics. All authors have read and agreed to the published version of the manuscript.

**Funding:** This paper was part of the Ph.D. research funded by the Intra-Africa Academic Mobility Scheme of the European Union (EU).

**Acknowledgments:** We acknowledge the support from the EU for funding the research. The authors would also like to thank students and staff of the University of Namibia who played critical roles in logistics and actual data collection. Staff of the Ministry of Agriculture, Forestry and Water Development from Omusati, Oshana, Ohangwena, Oshikoto and Kavango West regions, and town council staff from Outapi, Oshakati and Ongwediva town councils are also acknowledged for providing logistical support during data collection.

**Conflicts of Interest:** The authors declare no conflict of interest.

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
