# Peer review of "Diversity and Domestication Status of Spider Plant (Gynandropsis gynandra, L.) amongst Sociolinguistic Groups of Northern Namibia"

_agronomy, doi:10.3390/agronomy10010056_

Round 1

Reviewer 1 Report

General comments

In this work was investigated the species diversity and domestication of spider plant (Cleome gynandra) variants, in a way to impact in pre-breeding for client-preferred traits in Namibia.

The authors concluded that the farmers in the study regions of Namibia does not active domesticated spider plants. The farmers do not perform a transplanting of spider plants from the wild into fields and gardens, a selection of seeds of specific plant variants for reproduction or try to control its pests and diseases. Exist a variation in the level of attention paid by farmers from different sociolinguistic groups to spider plants.

In these regions of Namibia seems naturally (without selection) occur plants with green stem, green petiole morphotypes, with and without trichomes, which in comparison to purple accessions could be less nutritive, healthier and presents less phytochemicals which confer resistance to insect pests. The tight diversity status of spider plants in the study regions limit like this an implementation of breeding program’s in north of Namibia.

In my opinion the information obtained in this work is very interesting, innovative and arise knowledge with a high importance for the future application of this species in nutrition of population in poor regions of sub-Saharan Africa. However, this study is only the first step for the implementation of breeding program’s in Namibia. Other regions of Namibia need to be study in future works. Breeding studies need to be conducted in restricted green house conditions, regarding the cross between native spider plants and purple stem spider plants that was already described with beneficial nutritional and healthier traits.

Specific comments

Line 34- I substitute “which is” by “that is”.

Line 41 – Substitute “value of spider plant” by “value of the spider plant”.

Line 43 – Substitute “for domestication” by “for the domestication”.

Line 46 – Substitute “as relish, sold in the market as green or processed vegetable” by “as a relish, sold in the market as a green or processed vegetable”.

Line 47 – Substitute “in dry season” by “in the dry season”.

Lines 47 to 48 – Substitute “plant either by sun drying or 47 bleaching followed by drying or forming of pellets which are used as relish” by “plants either by sun-drying or 47 bleaching’s followed by drying or forming of pellets which are used as a relish”.

Line 56 – Substitute “, introduction” by “, the introduction”.

Line 65 – Substitute “recommended ethnobotanical” by “recommended an ethnobotanical”.

In methodology, I feel that miss a table with the agro-morphological features of the spider plants considered in the interviews.

Line 171 to 172 – I do not understand how the information described in these sentences are present in figure 2.

Line 153 – Substitute “Mean…” by “The mean…”.

Line 161- Substitute “In literature, spider..” by “In literature, the spider….”.

Line 167 – Substitute “globulous spider plant while 52%” by “globulous spider plants while 52%..”.

Line 169 – Substitute “Abundance of the spider plant..” by “The abundance of the spider plants….”.

Line 184 – Substitute “domestication of spider plant” by “domestication of spider plants”.

Line 187 – Substitute “translated…” by “translate”.

Line 191 – substitute “including selection” by “including the selection”.

Line 208 to 209 – Substitute “There was correlation in domestication trends between the location of sociolinguistic group and the regions.” by “There was the correlation in domestication trends between the location of the sociolinguistic group and the regions”.

Line 220 - Substitute “study showed lack” by “study showed a lack..”.

Line 225 - Substitute “towards…” for “to”.

Line 226 - Substitute “through enhancement” by “through the enhancement”.

Line 234 – Substitute “which did not undergo..” by “that did not undergo..”.

Author Response

Please find uploaded my response to reviewer 1. All the comments have been addressed. 

Reviewer 2 Report

The manuscript entitled “Diversity and domestication status of spider plant (Gynandropsis gynandra, L.) amongst sociolinguistic groups of northern Namibia” under consideration for publication in Agronomy, evaluates diversity, occurrence and domestication levels of Gynandropsis gynandra  in northern Namibia.

The research presented in the manuscript based on semi-structured interviews involving 100 randomly selected farming households, four key informant interviews and one focus group discussion.

Interviews are a very good and detailed source of knowledge about the occurrence and spread of flora and fauna species. Using them, we can determine the scale of threatened with extinction, expansion or domestication of many plant species. The test results presented in the manuscript have a high educational value.

This results indicate, that in northern Namibia, diversity of the species was limited to two morphotypes with green petiole and green stem and  this species is still in the early stages of domestication. The numbers of spider plant are decreasing due to soil fertility declining.

The manuscript is very interesting, but needs to improve and systematize. Therefore, I propose:

In the manuscript there is missing citation of publications from no 26 to 62. The authors cited only publications up to no 25.

In the manuscript, the authors cited on the first place publication no 25, on the second place publication no 24. Please change the references order, because it has to be written in the same order they are cited. Lines 161-161 The authors wrote "In literature, spider plant is categorized into four morphotypes based on pigmentation, and these are green, purple, pink and violet", but there is no reference in literature. Please provide it with the relevant publications.

In the section " 3.2 Diversity of Spider plant " to add a table or diagram with the occurrence of individual spider plant morphotypes in studied regions and ethnic groups of northern Namibia. There is incorrect numbering of figures in the manuscript: Lines 171 and 172 to change figure 2 to figure 3 Line 196 to change figure 3 to figure 4 Line 204 to change figure 4 to figure 5 Line 204 to change figure 5 to figure 6 In the figure 5 and 6 to change „Domentication” to „Domestication”.

There are numerous repetitions and similar statements in the manuscript, e.g. lines 53-55 „Studies have suggested that utilizing genetic diversity effectively is key towards achieving the gains in production and productivity [3] and this provides the basis for enhancing plant domestication and genetic improvement” and lines 225-226 „Genetic diversity has been reported to be the key towards achieving the gains in production and productivity [3] through enhancement of plant domestication and genetic improvement. Please analyze the manuscript once again, according to the recommendations and remove repetition contents.

Author Response

Please find uploaded my response to reviewer 2. All the comments have been addressed.
